# Brain-Derived Neurotrophic Factor in Gestational Diabetes: Analysis of Maternal Serum and Cord Blood Pairs and Comparison of Dietary- and Insulin-Dependent GDM

**DOI:** 10.3390/metabo12060482

**Published:** 2022-05-26

**Authors:** Michael Robert Jaskolski, Anna Katharina Diedrich, Alexandru Odainic, Susanne Viktoria Schmidt, Marie-Therese Schmitz, Brigitte Strizek, Ulrich Gembruch, Waltraut Maria Merz, Anne Flöck

**Affiliations:** 1Department of Obstetrics and Prenatal Medicine, University Bonn Medical School, 53127 Bonn, Germany; michael.jaskolski@ukbonn.de (M.R.J.); anna_katharina.diedrich@ukbonn.de (A.K.D.); brigitte.strizek@ukbonn.de (B.S.); ulrich.gembruch@ukbonn.de (U.G.); waltraut.merz@ukbonn.de (W.M.M.); 2Institute of Innate Immunity, University Bonn Medical School, 53127 Bonn, Germany; aodainic7@gmail.com (A.O.); susanne.schmidt@uni-bonn.de (S.V.S.); 3Department of Microbiology and Immunology, The Peter Doherty Institute for Infection and Immunity, University of Melbourne, Melbourne, VIC 3000, Australia; 4Department of Medical Biometry, Informatics and Epidemiology, University Bonn Medical School, 53127 Bonn, Germany; m.schmitz@imbie.uni-bonn.de

**Keywords:** brain-derived neurotrophic factor, cord blood, cytokines, gestational diabetes, inflammation, obesity, placenta, pregnancy

## Abstract

The Objective of our study was to investigate the influence of dietary (dGDM) and insulin-dependent (iGDM) gestational diabetes (GDM) on BDNF blood levels of corresponding maternal-neonatal pairs and compare them to pregnancies unaffected by GDM. Blood samples from 293 maternal-neonatal pairs were analyzed. Statistical analysis was performed using multiple regression analysis for association of log-transformed maternal and neonatal BDNF levels in relation to GDM, gestational age, neonatal sex, and mode of delivery. This was followed by a 2:1 matching of healthy and diabetic pairs. Maternal and neonatal BDNF levels were lowest in the iGDM group, followed by the dGDM group and healthy controls (maternal: healthy 665 ± 562 (26–2343) pg/mL vs. dGDM 593 ± 446 (25–1522) pg/mL vs. iGDM 541 ± 446 (68–2184) pg/mL; neonate: healthy 541 ± 464 (9.5–2802) pg/mL vs. dGDM 375 ± 342 (1–1491) pg/mL vs. iGDM 330 ± 326 (47–1384) pg/mL). After multiple regression analysis and additional 2:1 matching neonatal log-BDNF was significantly lower (−152.05 pg/mL, *p* = 0.027) in neonates of mothers with GDM compared to healthy pairs; maternal log-BDNF was also lower (−79.6 pg/mL), but did not reach significance. Our study is the first to analyze BDNF in matched maternal-neonatal pairs of GDM patients compared to a metabolically unaffected control group.

## 1. Introduction

Gestational diabetes (GDM) is widespread, with rates increasing during recent years and a prevalence of 25.5% in certain populations [1,2,3]. It is associated with an increased risk of several complications during pregnancy and long-term consequences for mother and child [4]. People with diabetes mellitus (DM) and children of diabetic mothers have a higher risk of impaired brain function and psychiatric disorders [5,6] as GDM affects fetal neurodevelopment [7].

Known to be protective for neuronal survival and reconstruction after damage, the neurotrophin BDNF [8,9] has recently been introduced as metabotrophic factor involved in energy expenditure, lipid and glucose level control, and cardiovascular homeostasis [10,11], as well as fetal development [12,13,14,15,16]. BDNF has been described to be lower in diabetic patients [17,18] and with diabetic complications [19,20], but data are conflicting [5,21]. In experimental and animal studies, it has been shown that an infusion of BDNF in obese diabetic mice reduced the blood glucose level [11], a fact that has brought BDNF into focus as a treatment option in the future [22,23].

Data on the effects of glucose tolerance disorders on maternal and fetal BDNF levels during pregnancy are rudimentary. BDNF has been described to be influenced by a number of factors, such as gestational age, fetal sex, and mode of delivery [24,25,26].

The aim of our study was to investigate the influence of GDM on BDNF blood levels of corresponding maternal-neonatal pairs in singleton pregnancies in accordance with gestational age, neonatal sex, and mode of delivery.

## 2. Results

### 2.1. Baseline Characteristics

Two hundred and ninety-three (*n* = 293) maternal-neonatal pairs were recruited. In all, sixty (*n* = 60) pairs were excluded: twelve (*n* = 12) pairs due to twin pregnancies, *n* = 12 due to fetal anomalies, *n* = 9 due to preeclampsia, and *n* = 26 due to maternal diseases that could affect maternal BDNF (psychiatric disorders *n* = 19, neurological disorders *n* = 7). One pair was excluded due to known pre-existing DM type II, leaving 233 mother blood-cord blood pairs for analysis. *n* = 173 (74.3%) pairs were healthy, in *n* = 60 (25.8%) the mother was diagnosed with GDM. Of those, *n* = 31 (13.3%) were classified as iGDM; in *n* = 29 (12.5%) glycemic control was sufficient with diet (dGDM) (see Figure 1). Demographic and obstetric data are listed in Table 1.

### 2.2. Comparisons between Groups

**(1)** 
**Maternal and neonatal BDNF: healthy (*n* = 173) vs. all GDM (*n* = 60)**


Mean maternal and neonatal BDNF values (±SD) were lower in the GDM group compared to healthy controls (maternal: GDM 566.2 ± 443.0 (range 24.6–2184.0) pg/mL vs. healthy 664.6 ± 562.4 (range 25.7–2343.0) pg/mL; neonatal: GDM 351.7 ± 331.8 (range 1.0–1490.0) pg/mL vs. healthy 541.3 ± 463.9 (range 9.5–2802.0) pg/mL).

For neonatal BDNF, a significant difference was present between newborns of diabetic mothers compared to healthy controls (*p* = 0.001), while maternal BDNF levels showed no difference (*p* = 0.406).

**(2)** 
**Maternal and neonatal BDNF: healthy (A; *n* = 173) vs. dGDM (B; *n* = 29) vs. iGDM (C; *n* = 31)**


Mean maternal and neonatal BDNF values were lowest in the iGDM group, followed by the dGDM group and healthy controls (maternal: healthy 664.6 ± 562.4 (range 25.7–2343.0) pg/mL vs. dGDM 593.1 ± 446.1 (range 24.7–1522.1) pg/mL vs. iGDM 541.0 ± 446.1 (range 67.7–2184.0) pg/mL; neonatal: healthy 541.3 ± 463.9 (range 9.5–2802.0) pg/mL vs. dGDM 374.6 ± 342.0 (range 1.0–1491.0) pg/mL vs. iGDM 330.2 ± 326.3 (range 46.6–1384.0) pg/mL).

The differences in neonatal BDNF levels reached significance level both between healthy vs. iGDM (*p* = 0.003) and healthy vs. dGDM (*p* = 0.042). Comparison between the iGDM with dGDM groups showed no significant differences in neonatal BDNF levels (*p* = 0.394) (see Figure 2). 

Evaluation of maternal BDNF levels showed no significance between the groups (*p* = 0.652).

### 2.3. Multiple Regression Analysis

After adjusting for gestational age, neonatal sex, and mode of delivery, maternal log-BDNF did not significantly differ between healthy mothers and those with GDM (*p* = 0.554).

Contrasting to that, neonatal log-BDNF was significantly associated between newborns of healthy mothers and those with GDM (*p* = 0.007).

### 2.4. Matching

After matching for gestational age, neonatal sex, and mode of delivery, maternal log-BDNF was not significantly lower (−79.6 pg/mL, *p* = 0.34, see Figure 3) in mothers with GDM. However, the neonatal log-BDNF was significantly lower (−152.05 pg/mL, *p* = 0.027, see Figure 4) in newborns of mothers with GDM.

## 3. Discussion

BDNF is widely described as a factor involved in multiple cerebral processes in adult life with higher levels assumed to be protective for several diseases such as depression or Alzheimer‘s disease [27,28]. It has, furthermore, been introduced as a metabotrophic factor involved in energy expenditure, and lipid and glucose level control [8,9]. Given the facts that DM affects maternal neurophysiology and fetal neurodevelopment [5,6,7] and animal studies showed a reduction of blood glucose levels after BDNF infusion [11], the question arises whether fetal BDNF and possibly fetal BDNF-guided neuronal growth, differentiation, and synaptogenesis could be influenced by different prenatal glycemic control.

Our study is the first to analyze matched mother–newborn pairs between healthy controls and patients with dGDM and iGDM. We were able to show that after matching of maternal and neonatal variables 1:2 for gestational age, neonatal sex and mode of delivery maternal log-BDNF was lower in diabetic mothers, but did not show statistical significance (*p* = 0.34). However, the diabetic neonatal log-BDNF was significantly lower than in healthy controls.

### 3.1. BDNF in Diabetes Mellitus

Our findings are in line with other studies showing lower levels of blood BDNF in non-pregnant women with DM type II compared to healthy controls [17,18]. Interestingly, Fujinami et al. [17] were able to show an inverse correlation between duration of diabetes and BDNF levels. However, data on BDNF in diabetic patients are conflicting, a fact that could be attributed to the heterogeneity of groups being analyzed. Boyuk et al. [21] and Suwa et al. [29] found higher BDNF levels in diabetic patients compared to healthy controls, whereas Lee et al. [30] reported on similar levels between adolescents with DM type II and controls.

### 3.2. BDNF in GDM

Studies on the influence of GDM on fetal and maternal blood BDNF levels are rare, characterized by small numbers, or analyzed by either mother or fetus or placenta [28,29,30,31,32,33,34,35]. Our data are complementary to previously published results and show an inverse correlation between hyperglycemia and BDNF concentration. This would also explain the different significance levels between dGDM and iGDM, as BDNF was found to be decreased in dGDM compared to the control group, but the difference was clearly more significant in the iGDM group. This explanation is based on the assumption that in iGDM the dietary therapy had to be expanded with insulin due to inadequate blood glucose control and, thus, an overall more severe course of the disease with higher fasting glucose levels may be assumed [4]. A strong negative correlation has been described between fasting glucose levels and BDNF blood levels [10], and insulin has a direct effect on fasting glucose. Interestingly, Sardar et al. [36] concluded that gestational diabetes disrupts BDNF expression in the hippocampus of rat pups and insulin was sufficient to prevent these alterations. A fact that may help to explain the affection of fetal neurodevelopment during GDM [7].

Energy homeostasis is controlled by different complex mechanisms, including hormonal signaling and multiple molecules. Increasing studies shed light on synergies between signaling pathways of molecules involved simultaneously in appetite control, energy regulation, reproduction, and neuroscience [5,10,37,38]. The complexity of connections between different regulation axes is the main problem that hampers the applicability of possible treatments. As an example, on the one hand, hypothalamic anorexigenic and orexigenic effects of BDNF, RVD-hemopressin α, and kisspeptin showed possible treatment options in animal models [11,22,23,38,39]; on the other hand, anxiogenic effects of hemopressin administration were observed in animals. 

### 3.3. Neonatal BDNF in Umbilical Cord Blood

In 2017, Cai et al. [31] showed a significant down-regulation of BDNF in *n* = 58 non-diabetic large for gestational age (LGA) fetuses compared to a healthy appropriate for gestational age (AGA) control group analyzed in cord blood as well as placental samples and postulated a protective effect of BDNF against non-diabetic macrosomia. In 2018, Briana et al. [32] investigated BDNF levels in 80 cord-blood samples of pregnancies with GDM and different fetal growth patterns (FGR, LGA, and AGA) compared to an AGA control group of metabolically healthy pregnancies. Significantly lower BNDF levels were found in newborns of mothers with GDM, regardless of the intrauterine growth pattern. However, female fetuses had increased BDNF levels. Maternal BDNF levels were not examined and no distinction was made regarding GDM therapy. In our study, sex was, therefore, included into the matching of pairs.

Su et al. [34] published longitudinal data from a small group of *n* = 18 fetuses whose mothers were diagnosed with GDM in pregnancy compared to *n* = 16 healthy controls. Newborns of diabetic mothers tended to have lower serum BDNF compared to the control group, without reaching significance level. Maternal BDNF levels were not examined and no distinction was made with respect to GDM therapy.

In contrast to our findings, Bayman et al. [35] found higher BDNF levels in *n* = 20 cord blood samples of mothers with iGDM, but no difference between cord blood levels of all diabetic vs. non-diabetic samples (*n* = 43 vs. *n* = 53). The difference may be attributed to differences between the groups, e.g., BMI, degree of obesity, gestational age, mode of delivery, and maternal age.

### 3.4. Maternal BDNF

There are no studies analyzing blood levels in women with GDM compared to healthy controls. Jadhav et al. [33] found no significant differences in placental BDNF levels between *n* = 60 patients with GDM and *n* = 70 non-diabetic patients. Here, the significant difference in gestational age between both groups may have been a limiting factor.

### 3.5. Factors Contributing to BDNF Concentration

Complexity arises from the fact that a number of factors exert an influence on BDNF concentrations [24,25,26] and diagnostic criteria of GDM are different throughout the world [40]. Additionally, glycemic control, degree of systemic inflammation, and possibly preexisting diabetes in the selected patients may be influencing factors. Our study is the first that took these factors into consideration by matching maternal and newborn sample pairs according to possible influencing factors and comparing groups with different glycemic control.

## 4. Material and Methods

This cross-sectional, prospective study was performed between August 2019 and December 2020 in the obstetric unit of a tertiary referral center. The study was approved by the institutional ethics committee (number 305/11). All women gave their written informed consent. Baseline demographic and clinical data, obstetric history, and course of the current pregnancy including pregnancy complications, delivery and newborn data were obtained from the patients’ charts and the departmental database. Twin pregnancies, fetuses with known chromosomal abnormalities or major sonographic anomalies, and patients with preexisting DM type I or II or with maternal diseases that could affect maternal BDNF concentration like psychiatric or neurological disorders were excluded. Ultrasonographic and echocardiographic evaluation was performed only by experienced sonographers with high-resolution ultrasound equipment (Philips EpiQ7, Philips Hamburg, Germany; Voluson E10, GE Munich, Germany, Aplio 900, Canon Medical Systems, Neuss, Germany) in all cases, with 5–9 and 2–6, 7, and 8 MHz convex transducers, respectively. A detailed assessment of the fetal anatomy and cardiovascular status including echocardiography and Doppler examination was performed in all subjects. 

All patients had routine oral glucose tolerance testing at 24–28 weeks of gestation using a “two-step” approach and were classified as GDM according to the following criteria: (1) non-fasting 50 g oral glucose challenge test (GCT) 7.5–11.1 mmol/L (135–200 mg/dL) followed by (2) 75 g oral glucose tolerance testing (OGTT) with one or more than one glucose levels at fasting, 1 h, and 2 h post-75 g glucose load ≥5.1, 10.0 and 8.6 mmol/L (92, 180 and 153 mg/dl), respectively [41,42].

Maternal venous blood samples (6 mL) were taken on admission to the labor ward. Neonatal blood samples were taken as newborn venous blood samples (6 mL) from the placental part of the umbilical cord, immediately after clamping of the cord, and before delivery of the placenta. All samples were transferred to +4 °C until centrifugation (4000 rpm, 10 min); thereafter, serum was separated and stored at −80 °C.

BDNF concentrations in maternal and neonatal serum were quantified according to the manufacturer’s protocol with the human BDNF Simplex ProcartaPlex Kit (Thermo Fisher Scientific, Waltham, MA, USA). For all samples, no more than one freezing cycle was allowed. Defrosted samples were centrifuged again at 10,000× *g* at 4 °C for 10 min to remove the particulates. Fluorescence intensities for BDNF were analyzed via the FLEXMAP 3D System (Luminex, ’s-Hertogenbosch, The Netherlands). The mean fluorescence intensities of the standard dilution series were fitted to a 5-parameter linear weighted curve to determine the BDNF concentration of each serum sample and normalized by the Milliplex Analyte software (v5.1.0.0, VigeneTech, Inc., Carlisle, MA, USA). Limit of detection was 1.19 pg/mL.

For data analysis, the statistical software package R (2020; v4.0.2, R: A language and environment for statistical computing. R Foundation for Statistical Computing, Vienna, Austria. URL https://www.R-project.org/, accessed on 12 April 2022) and SPSS Statistics 28.0 (IBM, Armonk, NY, USA) were used. Between group comparisons of approximately normally distributed variables were performed by analysis of variance or *t*-tests for independent samples, otherwise Kruskal–Wallis or Mann–Whitney U Test was used. Chi-square tests were performed for categorical variables.

Multiple regression analysis was performed to evaluate the association of log-transformed maternal or neonatal BDNF and GDM adjusted for gestational age (weeks), neonatal sex, and mode of delivery. Additionally, as sensitivity analysis, the association between maternal or neonatal BDNF and GDM was evaluated using matched samples. The matching procedure based on one-to-two nearest-neighbor matching on the propensity score included the variables: gestational age (weeks), neonatal sex, and mode of delivery. The assessment of the effectiveness of the matching procedure was based on standardized means differences. A level of *p* < 0.05 was considered statistically significant.

## 5. Conclusions

Maternal and neonatal BDNF is lower in diabetic patients, but only shows statistical significance in the neonatal group compared to healthy controls. Further longitudinal human studies with predefined diagnostic criteria and follow-up are needed to gain a deeper understanding of the role of BDNF and glycemic control in the pre- and perinatal period.

### Limitations

A possible limitation arises from the fact that women with diagnosis of GDM may, in fact, have had undiagnosed preexisting DM. In our study, we excluded all women with known preexisting DM, and patients with known hyperglycemia in the first trimester were classified as presumed preexisting DM and were also excluded. Another limitation may be the different timepoints of GDM diagnosis and a lack of predefined criteria to distinguish patients with good and poor glycemic control.

Furthermore, assay-specificity is a general limitation which hampers comparisons of studies.

## Figures and Tables

**Figure 1 metabolites-12-00482-f001:**
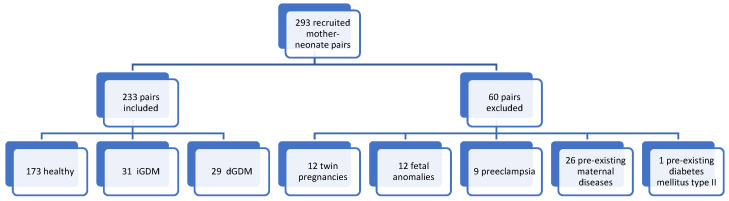
Flow diagram. Distribution of maternal–neonatal pairs by inclusion or exclusion with the reasons for exclusion. Furthermore, subdivision of the included pairs according to healthy, dietary- and insulin-dependent GDM.

**Figure 2 metabolites-12-00482-f002:**
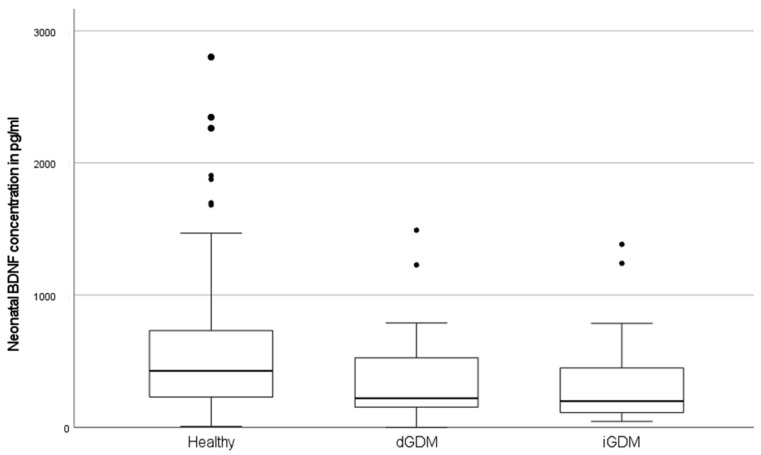
BDNF concentration (pg/mL) in neonates of mothers with dietary (dGDM), insulin-dependent gestational diabetes (iGDM) and metabolically healthy controls.

**Figure 3 metabolites-12-00482-f003:**
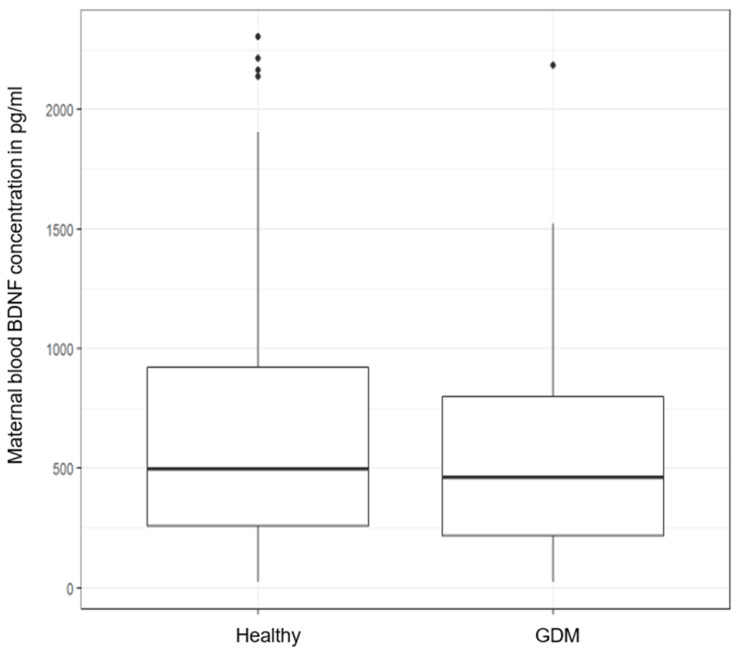
Maternal BDNF concentration (pg/mL) in cases with GDM and metabolically healthy controls (*p* = 0.34).

**Figure 4 metabolites-12-00482-f004:**
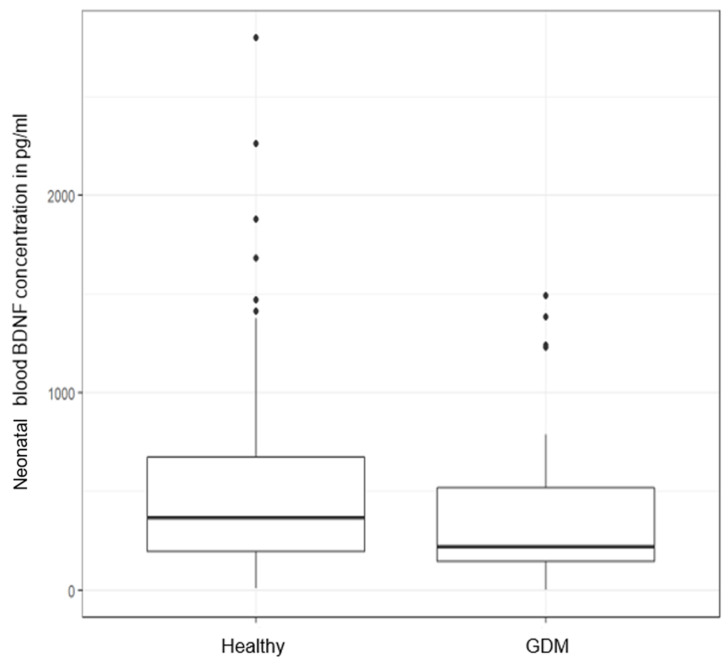
BDNF concentration (pg/mL) in neonates of mothers with GDM and metabolically healthy controls (*p* = 0.027).

**Table 1 metabolites-12-00482-t001:** Maternal, neonatal, and sample processing data of group A (healthy mothers), group B (dGDM), and group C (iGDM).

KERRYPNX	Healthy (*n* = 173)	dGDM (*n* = 29)	iGDM (*n* = 31)	*p*
**Maternal data:**				
**Age** (years)				0.0501
mean ± SD	33.0 ± 5.0	32.1 ± 5.7	35.2 ± 5.2	
**BMI before pregnancy** (kg/m^2^)				<0.001 A vs. B A vs. C
mean ± SD	26.5 ± 20.4	28.9 ± 8.1	32.7 ± 12.4	
**BMI before pregnancy, category** (kg/m²) % (*n*)				-
<30	92.5 (160)	65.5 (19)	54.8 (17)	
30–34.9	3.5 (6)	13.8 (4)	16.1 (5)	
>35	4.1 (7)	20.7 (6)	29.0 (9)	
**BMI at delivery** (kg/m²)				<0.001
mean ± SD	28.5 ± 4.9	33.0 ± 7.5	34.4 ± 8.7	
missing (*n*)	(1)	-	-	
**BMI at delivery, category** (kg/m²) % (*n*)				-
<30	68.6 (118)	41.4 (12)	38.7 (12)	
30–34.9	23.8 (41)	27.6 (8)	16.1 (5)	
>35	7.6 (13)	31.0 (9)	45.2 (14)	
missing (*n*)	(1)	-	-	
**Platelet count** (*n*/µL)				0.490
mean ± SD	223.1 ± 81.6	230.7 ± 49.3	230.9 ± 77.8	
missing (*n*)	(7)	(2)	-	
**Smoker** % (*n*)				
yes	- (0)	3.4 (1)	3.2 (1)	
unknown	10.4 (18)	13.8 (4)	9.7 (3)	0.124
**BDNF serum concentration** (pg/mL)				0.652
mean ± SD	664.6 ± 562.4	593.1 ± 446.1	541.0 ± 446.1	
**Neonatal data:**				
**BDNF serum concentration** (pg/mL)				0.004
mean ± SD	541.3 ± 463.9	374.6 ± 342.0	330.2 ± 326.3	A vs. B
missing (*n*)	(3)	-	-	A vs. C
**Gestational age at delivery**, (weeks)				
mean ± SD	39.7 ± 2.0	39.9 ± 1.2	39.0 ± 1.5	0.008
missing (*n*)	(1)	-	-	A vs. C
**Neonatal sex** % (*n*)				0.845
male	53.8 (93)	48.3 (14)	54.8 (17)	
female	46.2 (80)	51.7 (15)	45.2 (14)	
**Mode of delivery** % (*n*)				0.001
vaginal birth	59.3 (102)	75.0 (22)	25.8 (8)	
instrumental	5.8 (10)	3.6 (1)	9.7 (3)	
elective cesarean	27.9 (48)	14.3 (4)	61.3 (19)	
emergency cesarean	7.0 (12)	7.1 (2)	3.2 (1)	
missing (*n*)	(1)	(1)	-	
**Birth weight** (g)				0.559
mean ± SD	3392.4 ± 577.0	3352.3 ± 359.0	3496.8 ± 618.7	
**Birth weight percentile**				0.077
mean ± SD	50.4 ± 28.4	46.4 ± 24.9	61.5 ± 27.4	
**Umbilical artery cord pH**				0.804
mean ± SD	7.3 ± 0.1	7.3 ± 0.1	7.3 ± 0.1	

## Data Availability

Not applicable.

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
