# Peer review of "Brain-Derived Neurotrophic Factor in Gestational Diabetes: Analysis of Maternal Serum and Cord Blood Pairs and Comparison of Dietary- and Insulin-Dependent GDM"

_metabolites, 2022, doi:10.3390/metabo12060482_

Round 1

Reviewer 1 Report

The current paper represents original author contribution focused on influence of gestational diabetes (GDM) on BDNF blood levels of corresponding maternal-neonatal pairs. The study design was prospective and different from previously conducted in context of  maternal-neonatal pairs. If it is possible it might be interesting to add maternal metabolic parameters if they are available (maybe glucose values from 2h OGTT).

Author Response

  • Reviewer 1: “If it is possible it might be interesting to add maternal metabolic parameters….. (maybe glucose values from 2h OGTT)”

 Unfortunately, we cannot provide these data. Routine oral glucose tolerance testing at 24 - 28 weeks of gestation is non-uniform in Germany and in our patients. Test were done in resident doctors practices even with non-fasting 50g oral glucose challenge test followed by a 75g oral glucose tolerance testing (OGTT) if suspect, or only 75g OGTT, respectively

Reviewer 2 Report

Dear Author

The study is interesting and deserves consideration in this journal.

The english needs revision for typos and grammar.

I suggest to add a little discussion on the influence of hormones such as neuropeptides in brain derived fuctors and diabetes, for example kiss-peptin and RVD hemopressin; look at the following recent literature es example: "Effects of Kisspeptin-10 on Hypothalamic Neuropeptides and Neurotransmitters Involved in Appetite Control", "Anorexigenic effects induced by RVD-hemopressin(α) administration", "Emotional disorders induced by Hemopressin and RVD-hemopressin(α) administration in rats".

Author Response

  • Reviewer 2:

“The english needs revision for typos and grammar”

 The manuscript was revised by a native speaker, grammar and typing mistakes were corrected.

“I suggest to add a little discussion on the influence of hormones such as neuropeptides in brain derived functions and diabetes, for example kisspeptin and RVD-hemopressin; …”

 We added a short discussion on the topic in the discussion section 3.2 BDNF in GDM, see lines 240-255. Accordingly, we added four new reference (36-39).
